



# Dominant role of charged meteoric smoke particles in the polar mesospheric clouds

Liang Zhang[1], Zhongfang Liu[1], Brian Tinsley[2]

[1]State Key Laboratory of Marine Geology, Tongji University, Shanghai, 200092, China
[2]Physics Department, University of Texas at Dallas, Richardson, Texas, 75080, USA

*Correspondence to*: Liang Zhang (Liangzhang420@tongji.edu.cn)

**Abstract.** Polar mesospheric clouds (PMCs), composed of ice particles, are sensitive to solar activity and atmospheric dynamics, and have been suggested as a potential indicator of climate change. However, the microphysical processes of PMCs, especially the mechanism of ice nucleation, remain poorly understood. This study presents an analysis of satellite PMC data, which reveals that the mean ice particle radius ($r$) and concentration ($N$) in PMCs are primarily influenced by the PMC height ($h$), rather than by the mean environment temperature ($T$). Additionally, the ice particle column concentration ($N_c$) exhibits a surprising decrease with latitude. These results support the hypothesis that charged meteoric smoke particles (MSPs) act as ice nuclei, based on which we propose the charged-MSPs nucleation (CMN) scheme for the PMC formation. In contrast to the conventional growth-sedimentation (GS) scheme, in the CMN scheme the nucleation occurs throughout the PMC altitude range, ice particles grow mainly in situ, and the ice particle radius is determined by the competition for the limited water vapor rather than by sedimentation. The CMN scheme provides new pathways for solar activity and atmospheric dynamics to affect PMCs, and can explain a number of puzzling phenomena in the GS scheme.

## 1 Introduction

### 1.1 PMCs

Polar mesospheric clouds are the highest clouds in the Earth's atmosphere, usually forming in the polar mesopause when the temperature drops below 150 K in summer (Rapp and Thomas, 2006), and occasionally occurring in the mid-latitudes (Hervig *et al.*, 2016b; Miao *et al.*, 2022). Polar mesospheric clouds, together with polar mesospheric summer echoes (PMSEs) (Rapp and Lübken, 2004), both associated with ice particles, provide unique physical and dynamical information for the summer polar mesosphere. These ice particles are sensitive to perturbations in temperature and water vapor (Rong *et al.*, 2012), making PMCs susceptible to atmospheric dynamics, including gravity waves (GWs) (Chandran *et al.*, 2012; Gao *et al.*, 2018; Rusch *et al.*, 2017), planetary waves (France *et al.*, 2018; Liu *et al.*, 2015), tides (Fiedler and Baumgarten, 2018; Liu *et al.*, 2016), and inter-hemispheric coupling (IHC) (Gumbel and Karlsson, 2011; Karlsson *et al.*, 2009). The water vapor



released by rocket exhaust has been observed to enhance PMCs (Collins *et al.*, 2021; Stevens *et al.*, 2003), although the
long-term effects of space traffic on PMCs remains uncertain (DeLand and Thomas, 2019; Siskind *et al.*, 2013).

Solar ultraviolet (UV) radiation can heat the air and photolyze water molecules (Beig *et al.*, 2008; Shapiro *et al.*, 2012),
so it is straightforward to expect PMCs to be negatively correlated with the solar cycle. However, the 27-day solar cycle in
PMCs is typically ambiguous, unless the superposed epoch analysis method is applied to remove background noise, and the
lag time is only about 0-3 days (Dalin *et al.*, 2018; Robert *et al.*, 2010; Thurairajah *et al.*, 2017), shorter than the at least five
days required for the photodissociation (Shapiro *et al.*, 2012). The 11-year solar cycle in PMCs is evident prior to 2002, but
has disappeared over the past two decades, for reasons that remain unknown (Hervig *et al.*, 2019; Vellalassery *et al.*, 2023).
Global warming favors the formation of PMCs in two ways: increased carbon dioxide tends to cool the upper mesosphere
via the terrestrial radiation budget, and increased methane tends to increase the water vapor via oxidation (Lübken *et al.*,
2018). Nevertheless, the corresponding long-term trend in response to global warming has not been confirmed by
observations (Hervig *et al.*, 2016a; Kirkwood *et al.*, 2008), and whether the PMC can serve as an indicator of climate change
has been a topic of intense debate.

## 1.2 Nucleation processes in PMCs

The ice nucleation process contributes the most significant uncertainties to the microphysical process of PMCs.
Homogeneous nucleation is unlikely to occur at typical mesospheric supersaturation levels (Tanaka *et al.*, 2022), and
heterogeneous nucleation is thought to be responsible for the formation of ice particles by providing pre-existing ice nuclei,
for which the MSPs have been proposed as the most likely candidate (Duft *et al.*, 2019; Hervig *et al.*, 2012). Meteoric smoke
particles are produced by meteor ablation and recondensation and are abundant in the upper mesosphere, with radii ranging
from sub-nanometre to nanometre sizes (Plane, 2012; Scales and Mahmoudian, 2016).

The charged-MSPs have also been proposed as candidates for ice nuclei, given that the charges on MSPs can effectively
reduce the critical radius ($r_c$) at low temperatures (Gumbel and Megner, 2009; Megner and Gumbel, 2009). However, the
variability, distribution and charging process of MSPs remain unclear (Antonsen *et al.*, 2017; Dawkins *et al.*, 2023; Hervig *et
al.*, 2022; Knappmiller *et al.*, 2011; Li *et al.*, 2022), and this proposal has not yet gained widespread support (Duft *et al.*,
2019; Plane *et al.*, 2023). The mobility of electrons is much greater than that of positive ions, so MSPs in the upper
mesosphere tend to collect electrons, with rocket-borne measurements showing that about 10% of MSPs are negatively
charged (Plane *et al.*, 2014; Robertson *et al.*, 2014). It should be noted that the galactic cosmic rays can ionize the
troposphere and produce charged molecular clusters that grow much faster than neutral clusters, and the so-called ion-
mediated nucleation has long been proposed as an important way for the formation of cloud condensation nuclei in the
troposphere (Yu *et al.*, 2018; Yu and Turco, 2000).

The PMCs are located in the D-region of the ionosphere, where solar winds can exert an influence. The electric potential
of the ionosphere relative to the Earth's surface is about 250 kV, maintained by the global thunderstorms and electrified
clouds (Williams and Mareev, 2014). The dawn-dusk component of the interplanetary magnetic field (*IMF $B_y$*) can modulate



the polar ionospheric electric potential through the Lorentz electric field, $E_z = B_y \times V_x$, where the $V_x$ is the solar wind velocity. As the *IMF $B_y$* increases, the electric potential in the Southern Hemisphere (SH) polar region increases by ∼20 kV on average, thereby decreasing the electron density and the concentration of negatively charged-MSPs. In contrast, the response in the Northern Hemisphere (NH) is opposite (Lam and Tinsley, 2016; Tinsley and Heelis, 1993). Zhang *et al.* (2022) have demonstrated a positive/negative correlation between the *IMF $B_y$* and the ice particle radius of PMCs in the SH/NH with a lag time of about zero days, providing evidence in favor of the charged-MSPs acting as ice nuclei.

**1.3 The "growth-sedimentation" scheme for PMC formation**

The growth-sedimentation process is currently the most commonly used scheme to describe the formation of PMCs. In the GS scheme, ice nucleation occurs primarily at the top of the PMC, and then ice particles grow by water vapor deposition and fall downward under gravity, until they sediment out of the supersaturated zone and sublime (Hultgren and Gumbel, 2014; Rapp and Thomas, 2006). The lifetime of ice particles is also influenced by upward winds and diffusion. In the GS scenario, ice particle concentrations are higher at the top of the PMC and ice particle radii are larger at the bottom of the PMC due to ice particle growth and sedimentation, as supported by observations (Hervig et al., 2009a; Hultgren and Gumbel, 2014).

However, the conventional growth-sedimentation scheme for PMC formation faces a number of challenges. First of all, it seems unreasonable that ice nucleation takes place primarily at the top of the PMC, since the saturation levels throughout the PMC altitude range all favor PMC formation. Observed data indicates that the PMC is approximately in equilibrium with the surrounding environment, which is inconsistent with the dominant role of the sedimentation in the GS scheme (Christensen et al., 2016; Hervig et al., 2009c).

In the GS process, MSPs larger than the critical radius ($r_c \approx 1$ nm) are assumed to act as ice nuclei (Hervig *et al.*, 2012; Rapp and Thomas, 2006). However, the simulations involving mesospheric circulations showed that the large MSPs ($r > r_c$) are transported upwards along with the strong updrafts in the summer mesosphere, then transported into the winter mesosphere by the meridional winds, and finally sink to the stratosphere by downwelling (Megner et al., 2008a; Megner et al., 2008b). The global mass redistribution of MSPs results in a pronounced reduction of MSP concentration and lifetime in the summer mesosphere, thus there are not enough large MSPs in the summer polar mesosphere to act as ice nuclei.

In the GS process, the concentration of ice particle should decrease continuously from the top to the bottom of the PMC, but observations reveal the multi-layered structure of ice particles (Gao *et al.*, 2017; Kaifler *et al.*, 2013; Schäfer *et al.*, 2020). Moreover, the PMCs enhanced by the rocket exhaust plumes exhibit an inverse altitude dependence, with larger ice particles observed at higher altitudes (Stevens *et al.*, 2012). In the GS process, the entire life cycle of ice particles in PMC includes nucleation, growth, sedimentation and sublimation, which can take up to tens of hours (Rusch *et al.*, 2017; Wilms *et al.*, 2016), whereas PMCs vary more rapidly in observations (Megner *et al.*, 2018; Rusch *et al.*, 2009; Zasetsky *et al.*, 2009).

In the GS scenario, the PMC formation is thought to be controlled by temperature and water vapor, so a positive correlation between the ice particle radius and ice water content (*IWC*) is expected. However, this correlation is absent in



observations, for reasons that are still under debate (Megner, 2011; Megner, 2019; Wilms *et al.*, 2016; Wilms *et al.*, 2019).
What's more, satellite observations indicate a negative correlation between ice particle radius and albedo in the PMC regions affected by gravity waves (Gao *et al.*, 2018; Rusch *et al.*, 2017). Furthermore, the response of the ice particle radius in PMCs to the IMF $B_y$, as reported by Zhang et al. (2022), is also challenging to comprehend, given that the effects of the solar wind magnetic field are not incorporated into the GS scheme.

The PMC height is closely related to the fundamental processes of PMC formation, and is mainly determined by the
100 temperature structure in the upper mesosphere, with water vapor playing a minor role (Hervig *et al.*, 2009c; Russell III *et al.*, 2010). However, previous studies have focused on the effects of temperature and water vapor on the PMC formation, with little attention paid to the effect of PMC height. In this paper, we investigate the relationships for PMCs with altitude, temperature and latitude, based on which we propose the "charged-MSPs nucleation" scheme for PMC formation. The CMN scheme overcomes the limitations of the GS scheme, and provides new pathways to explain the response of PMCs to solar
activity and atmospheric dynamics.

## 2 Data and results

### 2.1 The PMC data

The PMC data applied in this work are observed by the Solar Occultation for Ice Experiment (SOFIE) and Cloud Imaging and Particle Size (CIPS) instruments on board the Aeronomy of Ice in the Mesosphere (AIM) satellite. Launched on 25 April
2007, the AIM satellite follows a sun-synchronous polar orbit and observes the PMC region approximately 15 times per day (Russell III *et al.*, 2009). The SOFIE platform performs occultation measurements to obtain vertical profiles of PMC properties, as well as the surrounding temperature and water vapor (Gordley *et al.*, 2009; Hervig *et al.*, 2009b). SOFIE covers regions at latitudes varying between 65° and 82° over time, and observes mainly around 70° latitude during the PMC season. The SOFIE provides polar mesospheric cloud data for 8 PMC seasons (from 2007 to 2014) in NH and 7 PMC
seasons (from 2007/2008 to 2013/2014) in SH, after which the SOFIE observes lower latitudes without PMCs. The CIPS platform consists of a panoramic UV nadir imager that observes the scattered radiance from PMCs, and images the PMCs in the 40°-85° latitude range with a horizontal spatial resolution of 5×5 km. The CIPS provides rectangular images of PMC properties in terms of ice particle radius, albedo, and *IWC* (Carstens *et al.*, 2013; Lumpe *et al.*, 2013). The CIPS data are available for 10 PMC seasons (from 2007 to 2016) in NH and 10 PMC seasons (from 2007/2008 to 2016/2017) in SH.

The SOFIE/AIM provides measurements for the altitude of the top ($Z_{top}$), maximum ($Z_{max}$) and bottom ($Z_{bot}$) of the ice layer in PMCs, as well as the altitude of the mesopause, as shown in Figure 1. The more frequent planetary waves in the NH cause stronger upward winds in the upper mesosphere, which adiabatically reduce the temperature and lead to an expansion of the saturation zone. Consequently, the altitude of the PMC bottom ($Z_{bot}$) observed by the SOFIE/AIM at about 70° latitude is approximately 1.7 km lower in the NH than in the SH, as shown in Fig. 1 and Table 1. The daily mean PMC height is
calculated by averaging the altitudes of the top and bottom of the PMC, $h = (Z_{top} + Z_{bot})/2$. The daily mean PMC thickness is



defined as the distance between the top and bottom of the PMC, $\Delta Z = Z_{top} - Z_{bot}$. The daily mean ice particle radius $r$ and concentration $N$ are calculated by averaging the ice particle radius and concentration inside the altitude range between the daily mean $Z_{bot}$ and $Z_{top}$.

## 2.2 Effects of altitude

Figures 2 and 3 demonstrate a significant negative correlation between the anomaly of daily mean ice particle radius $r$ and PMC height $h$ in both the NH and SH. The data anomalies are obtained by removing the 35-day running mean, and the time span of -10 to 50 days from the solstice is applied to define the PMC season. Similarly, Figures 4 and 5 demonstrate significant positive correlations for the daily mean ice particle concentration $N$ and the PMC height $h$. The mean sensitivity of $r$ to $h$ is about -3.2 nm/km in the NH and -2.6 nm/km in the SH, and the mean sensitivity of $N$ to $h$ is about 72.3 cm$^{-3}$/km

in the NH and 78.6 cm$^{-3}$/km in the SH. Figure 6 compares the mean correlation coefficients of the anomaly of $r$ and $N$ with $h$ in all PMC seasons to that of the anomaly of $r$ and $N$ with $\Delta Z$. In the GS scheme, both the lifetime of the falling ice particles in the supersaturated zone and the ice particle radius are expected to increase with increasing PMC thickness $\Delta Z$. Meanwhile, the mean ice particle radius $r$ and concentration $N$ are not expected to vary with PMC height $h$ in the GS scheme. In conclusion, the results presented in Fig. 6 demonstrate that the $r$ and $N$ are sensitive to $h$ rather than $\Delta Z$, which does not

support the GS scheme.

Figure 7 provides further insight into the zero-day lag time of the relationships for the ice particle radius $r$ and concentration $N$ with the PMC height $h$. In order to compare the results of CIPS and SOFIE, the variations in the ice particle radius and albedo observed by CIPS/AIM within the latitudes of 65° and 75° with respect to the PMC height $h$ observed by SOFIE/AIM at about 70° are calculated and presented in Figure 8, with the 3-day smoothing applied. Fig. 8 shows that the

145 ice particle radius $r$ observed by CIPS/AIM is also negatively correlated with $h$, in agreement with that of SOFIE. It is unexpected that the albedo observed by CIPS/AIM is also negatively correlated with $h$, which may be attributed to that some visible ice particles becoming subvisible as the mean ice particle radius $r$ decreases with increasing PMC height $h$.

## 2.3 Effects of temperature and water vapor

To investigate the influence of the surrounding environment on the PMC, the mean environment temperature $T$ and water

vapor $H_2O$ between 78 km and 88 km are calculated. Figure 9 (a) displays that the mean environment temperature $T$ mainly affects the PMC thickness $\Delta Z$, but not affect the PMC height $h$ and the ice particle radius $r$ and concentration $N$. Figure 10 shows the correlation coefficients for $Z_{bot}$ and $Z_{top}$ with temperature and water vapor at different altitudes, providing a more detailed illustration of the relationships between $h$ and $\Delta Z$ with the environment. Since the $Z_{top}$ is negatively correlated with the temperature at higher altitudes (>84 km), and $Z_{bot}$ is positively correlated with the temperature at lower altitudes (<82

155 km), as shown in Fig. 10 (a), the net effect is that the mean environment temperature $T$ mainly influences $\Delta Z$ but not $h$.



Figure 11 shows that the *IWC* observed by SOFIE/AIM is not affected by the PMC height *h*, but is positively correlated with the mean environment *T* and the PMC thickness $\Delta Z$.

The relationship between PMCs and water vapor is quite complex and ambiguous. The formation of ice particles in PMCs consumes water vapor, as illustrated by the negative correlation between *IWC* and $H_2O$ in Fig. 11. The freeze-drying effect influences the distribution of wate vapor by transporting it to lower altitudes through sedimentation and sublimation (Hervig *et al.*, 2015; Lübken *et al.*, 2009). For higher PMC bottoms, the freeze-drying effect may be less pronounced due to the smaller ice particle radii and weaker sedimentation, leading to the slightly positive correlation between the $Z_{bot}$ and the $H_2O$ at higher altitudes in the SH shown in Fig. 10 (b). Variations in water vapor may in turn affect the PMC height, i.e., when more water vapor is transported to the PMC top by upward winds, the $Z_{top}$ will increase accordingly, as shown by the slightly positive correlation between the mean environment water vapor $H_2O$ and the PMC height *h* in Fig. 9 (b).

In summary, the mean environment temperature *T* exerts a significant influence on the PMC thickness $\Delta Z$ and *IWC*, whereas the mean environment temperature *T* and water vapor $H_2O$ do not affect the ice particle radius *r* and concentration *N*. In the GS process, the formation of PMCs is generally thought to be controlled by the temperature and water vapor, so the results here are difficult to explain by the GS scheme.

## 2.4 Effects of latitude

Figure 12 shows the variations in *IWC*, ice particle radius, and ice particle column concentration obtained by CIPS/AIM with latitudes varying from 60° to 85°. The ice particle column concentration $N_c$ is estimated by dividing the *IWC* by the ice particle mass, $N_c = IWC/M_{ice}$, where the ice particle mass is estimated from the ice particle radius *r* and the ice density ($\rho_{ice} \approx$ 0.92 g cm$^{-3}$), $M_{ice} = 4\pi r^3 \rho_{ice}/3$. The PMC season in the CIPS/AIM data is defined as the time span from 0 to 40 days after the solstice, and the error bars represent the standard deviation of the means of the 10 PMC seasons.

It is well known that higher latitudes favor the formation of PMCs due to the stronger upwelling. As shown in Fig. 1 and Table 1, the thickness of PMCs is larger in the NH due to the stronger atmospheric disturbances there. Fig. 12 (a) and (b) demonstrate that the *IWC* and the ice particle radius *r* increase with latitude for both hemispheres, as expected. However, it is surprising that the ice particle column concentration $N_c$ decreases with latitude in the SH, as shown in Fig. 12 (c). In the GS scenario, the lifetime and concentration of the falling ice particles should increase with the PMC thickness and thus with the latitude. In brief, the decrease of ice particle column concentration $N_c$ with latitude in the SH poses another puzzle for the GS scheme.





## 3 Discussion

### 3.1 The "charged-MSPs nucleation" scheme for PMC formation

In order to elucidate the aforementioned results of PMC variations with altitude, temperature and latitude, we propose the "charged-MSPs nucleation" scheme for PMC formation, as illustrated in Figure 13. In the CMN scheme, the charged-MSPs act as ice nuclei by attracting water molecules through electric force, and ice nucleation occurs throughout the supersaturated region. The PMC is approximately in equilibrium with the surrounding environment, namely, the ice particles grow mainly in situ, with sedimentation playing a minor role. The ice particles continue to grow until the environment changes from

supersaturated to unsaturated due to the consumption of the local water vapor.

        According to the thermodynamic equilibrium, the mass of ice ($Q_{ice}$) at a given altitude is roughly equal to the difference between the total water vapor content ($Q_{tot}$) and the saturated water vapor content ($Q_{sat}$). The ice particle size is determined by the competition for the limited water vapor ($\approx Q_{tot} - Q_{sat}$), so the ice particle radius $r$ should be negatively correlated with the ice particle concentration $N$ at any altitude, as shown in Figure 14. Previous observations and simulations attribute the

negative correlation between $r$ and $N$ at the $Z_{max}$ to the competition for limited water vapor (Hervig *et al.*, 2009b; Hultgren and Gumbel, 2014; Wilms *et al.*, 2016). Fig. 14 extends this idea to each altitude, and supports the hypothesis that there is an excess of small charged-MSPs to act as ice nuclei for the entire PMC range. In the CMN scenario, a significant number of small ice particles in PMCs may be subvisible, providing an alternative explanation for the ice mass density estimated by thermodynamic equilibrium being higher than observed (Christensen *et al.*, 2016; Hervig *et al.*, 2009c).

The concentration of charged-MSPs increases rapidly with altitude, in accordance with the exponential increase in electron density produced by solar UV ionization. Therefore, the ice particle concentration $N$ and radius $r$ are mainly determined by the PMC height $h$, as shown in Fig. 6. For higher PMCs, more charged-MSPs are involved to act as ice nuclei and compete for the limited water vapor, resulting in smaller ice particles. Conversely, when PMCs occur at lower altitudes, fewer MSPs are charged by electrons, allowing ice particles to grow larger. This is consistent with the ground-based

observations that the lower PMCs are brighter (Gerding *et al.*, 2021; Ugolnikov and Maslov, 2019), possibly because the larger ice particles in the lower PMCs cause some subvisible ice particles to become visible.

        The mean environment temperature $T$ of the upper mesosphere between 78 km and 88 km mainly affects the PMC thickness $\Delta Z$ and $IWC$, but plays a minor role in the ice particle concentration $N$ and radius $r$, as illustrated in Fig. 9 and Fig. 11. The temperature at higher altitudes (>84 km) exerts a negative influence on the $Z_{top}$, while the temperature at lower

altitudes (<82 km) exerts a positive influence on the $Z_{bot}$, as shown in Fig. 10 (a). Consequently, the net effect is that the mean environment temperature $T$ primarily affects $\Delta Z$, but not $h$. In other words, the PMC height $h$ is influenced by the temperature structure rather than the mean temperature of the upper mesosphere.

        It is worth noting that Hervig *et al.* (2009c) identified a negative correlation between the temperature $T$ and ice particle concentration $N$ at $Z_{max}$. However, we argue that this correlation does not imply causality. Rather, from the perspective of the



CMN scenario, it is due to the opposite response of $T$ and $N$ to PMC height $h$. On the one hand, temperature and water vapor decrease dramatically with altitude in the upper mesosphere, therefore, both the mean temperature $T$ and water vapor measured within the PMC altitude range become smaller for higher PMCs, as evidenced by the negative correlations for $T$ with $h$ and $H_2O$ with $h$ shown in Figure 15. It should be noted that the mean temperature and water vapor at PMC altitude shown in Fig. 15 are obtained by averaging over the variable altitude range of the daily $Z_{bot}$ and $Z_{top}$, rather than the fixed

range of 78 km and 88 km applied in Fig. 9. On the other hand, the mean ice particle concentration $N$ is positively correlated with $h$, as shown in Fig. 6. In light of the aforementioned logic, it can be further argued that the positive correlation between $r$ and $H_2O$ at $Z_{max}$ reported by Hervig *et al.* (2009a), as well as the positive correlation between $r$ and $T$ at $Z_{max}$ reported by Hervig *et al.* (2009c), also do not represent causal relationships. Rather, they result from the responses of these variables to the PMC height $h$ variations.

The unexpected decrease in the ice particle column concentration $N_c$ with latitude, as shown in Fig. 12 (c), can be well explained in the CMN scenario. At higher latitudes, the stronger upward winds cause the $Z_{max}$ to be lower, as illustrated in Fig. 1. This results in fewer charged-MSPs being encountered, which in turn leads to fewer ice particles being formed at the lower $Z_{max}$. The CIPS/AIM measures PMCs primarily through the radiation reflected near $Z_{max}$. As a result, the ice particle column concentration obtained by CIPS/AIM decreases with latitude in the SH. In the NH, perturbations from lower

atmospheres are stronger, and the $Z_{max}$ observed by SOFIE/AIM near 70° is about 1.0 km lower, as shown in Fig. 1. Consequently, the reduction in $Z_{max}$ from 60° to 85° should be less pronounced in the NH, resulting in a minimal change in $N_c$ as depicted in Fig. 12 (c).

**3.2 Comparison of the CMN scheme with the GS scheme**

In the CMN scheme, the charged-MSPs act as ice nuclei, and the nucleation occurs throughout the PMC range. This is a

235 more reasonable scenario than the nucleation in the GS scheme, which occurs primarily at the top of the PMC. The concentration of MSPs varies exponentially with their radius, resulting in a greater number of small MSPs being available to attach electrons and act as ice nuclei in the CMN scenario, in contrast to the lack of large MSPs in the GS scenario due to meridional circulation transport.

The interpretation of the altitude distribution of ice particles in PMCs represents the primary distinction between the

240 CMN and GS schemes. In the CMN scenario, the observed increase in ice particle concentration $N$ with altitude is attributed to the altitude distribution of charged-MSPs, and the opposite distribution of ice particle radius $r$ is attributed to the competition for the limited water vapor, as illustrated in Fig. 13. In the GS scenario, the altitude distribution of $N$ and $r$ is primarily influenced by the growth and sedimentation of ice particles.

In contrast to the GS scenario, double-layer PMCs are permitted to form in the CMN scenario, provided that the

245 environment between the two layers is not conducive to PMC formation. In this case, ice particles in the upper and lower layers grow locally and independently. Furthermore, the mean ice particle radius in the lower layer is found to be larger than that in the upper layer (Gao *et al.*, 2017), which can be attributed to the fewer charged-MSPs in the lower PMCs. Stevens *et*



*al.* (2012) reported an inverse distribution of ice particle radius caused by rocket exhaust, which is unexplainable in the GS scenario. However, the CMN scenario offers a straightforward explanation for this phenomenon. The wate vapor injected by space shuttles above 100 km diffuses downwards, providing more water vapor to form ice particles at the top of the PMC, leading to the formation of larger ice particles above the smaller ice particles.

In the CMN scenario, ice nucleation occurs at each altitude, and all ice particles grow in situ simultaneously, allowing the PMC to form rapidly. This is evidenced by the zero-day lag time shown in Fig. 7. In addition, the charges on MSPs and on small ice particles, the presence of which has been demonstrated by PMSEs, are expected to increase the nucleation and growth rate, thereby contributing to the rapid changes in the PMC (Megner *et al.*, 2018; Rusch *et al.*, 2009; Zasetsky *et al.*, 2009). Nevertheless, the GS process requires a longer period for ice particles to grow and sediment over a distance of approximately 4 to 5 km from $Z_{top}$ to $Z_{bot}$ (Rusch *et al.*, 2017; Wilms *et al.*, 2016).

The lack of correlation between the ice particle radius $r$ and the *IWC* is puzzling in the GS scheme (Megner, 2011; Megner, 2019; Wilms *et al.*, 2016; Wilms *et al.*, 2019), but straightforward in the CMN scheme: the ice particle radius $r$ and concentration $N$ are sensitive to the PMC height $h$ rather than to the mean environment temperature $T$, while the *IWC* and the PMC thickness $\Delta Z$ are controlled by the mean environment temperature $T$, as shown in Figs. 6, 9, and 11.

## 3.3 New pathways for solar activity and atmospheric dynamics affecting PMCs

The CMN scheme provides a new pathway for the solar activity to affect PMCs via the charging process of MSPs. Zhang *et al.* (2022) report a positive response of the ice particle radius $r$, *IWC* and albedo to the IMF $B_y$ in the SH with a lag time of about zero days, and an opposite but weak response in the NH. These results are difficult to understand in the GS scheme, however, they are readily explicable in the CMN scheme: the increase in IMF $B_y$ induces an increase in the electric potential of the ionosphere in the SH, which reduces the concentration of electrons and the negatively charged-MSPs, ultimately leading to an increase in the ice particle radius. As the mean ice particle radius increases, some subvisible ice particles become visible, resulting in the positive response of the *IWC* and albedo of PMCs in the SH to the IMF $B_y$. In addition, the zero-day time lag of the PMC response to the IMF $B_y$ also confirms the rapid variation of the PMCs, as predicted by the CMN scheme.

Solar UV and IMF $B_y$ are usually not well correlated. During the solar minimum, the 27-day solar UV cycle tends to weaken, while the variations in IMF $B_y$ remain relative stable. Moreover, the charged-MSPs lag behind the IMF $B_y$ by about zero days, while water vapor lags behind the solar UV by at least five days. To summarize, the influence of IMF $B_y$ on PMCs does not align with that of solar UV, thereby contributing to the obscure 27-day solar signal in PMCs. In addition, solar UV can also impact the charging process of MSPs by ionizing the upper mesosphere. The lag time of charged-MSPs in response to solar UV is shorter than that of water vapor, which may explain why the observed lag time of PMCs in response to the 27-day solar UV cycle is as short as 0-3 days (Dalin *et al.*, 2018; Robert *et al.*, 2010; Thurairajah *et al.*, 2017).

The CMN scheme also provides a new pathway for atmospheric dynamics to influence PMCs via the PMC height. In the PMC regions affected by gravity waves, the ice particle radius $r$ is significantly negatively correlated with the PMC



brightness (Gao *et al.*, 2018; Rusch *et al.*, 2017), which is unexplainable in the GS scenario. However, this relationship can be well explained in the CMN scenario. During the cold phase of GWs, more water vapor is transported upwards. On the one hand, the variations in temperature $T$ and water vapor $H_2O$ favor the formation of PMCs, resulting in an increase in $\Delta Z$ and albedo. On the other hand, the PMC height $h$ increases as more water vapor is transported to the top of the PMC, as shown in

Fig. 9 (b), leading to the involvement of more charged-MSPs to act as ice nuclei, whose competition for the limited water vapor results in a decrease in the mean ice particle radius $r$. Similarly, during the warm phase of GWs, the PMC thickness $\Delta Z$ and albedo decrease due to the higher temperature $T$ and reduced water vapor $H_2O$, while the mean ice particle radius $r$ increases due to the lower PMC height $h$. In summary, as GWs pass through PMCs, both the PMC thickness $\Delta Z$ and PMC height $h$ change in response, causing the PMC brightness and ice particle radius to vary in opposite ways.

Inter-hemispheric coupling is induced by the sudden stratospheric warming (SSW) events in the winter hemisphere, causing the downward circulation anomaly in the summer polar mesopause which adiabatically heats the air and reduces the water vapor. During the IHC, similar to the warm phase of GWs, the brightness of PMCs should decrease, while the radius of ice particles should increase. SSWs are frequent in the NH but rare in the SH, so that the IHC is expected to lead to larger fluctuations in PMC height and ice particle radius in the SH than in the NH, as supported by Figs. 2-5 and Table 1.

## 295   4 Conclusion

This study demonstrates that the charged-MSPs nucleation scheme can explain a number of puzzles in the growth-sedimentation scheme and is likely to be responsible for the formation of PMCs. In the CMN scenario, ice nucleation occurs at all altitudes within the saturation zone, with charged-MSPs acting as ice nuclei. The concentration of charged-MSPs increases with altitude in accordance with the electron density, and their competition for water vapor determines the radius

of ice particles. The PMC exists approximately in equilibrium with the environment, i.e., ice particles grow mainly in situ, with sedimentation playing a minor role. The thickness $\Delta Z$, $IWC$, and albedo of the PMC are primarily influenced by the mean environment temperature $T$, while the ice particle concentration $N$ and radius $r$ are influenced by the PMC height $h$, which depends on the temperature structure in the upper mesosphere. For higher latitudes, the colder temperature results in the higher $IWC$ and larger ice particle radius $r$, but the lower PMC height $h$ results in fewer charged-MSPs and thus the

smaller ice particle column concentration $N_c$ in the SH.

     The CMN scenario provides a new pathway to understand the influence of atmospheric dynamics on PMCs. Specifically, the atmospheric disturbances can affect the PMC formation by modifying the PMC heights. This paper suggests that GWs and IHC can influence the PMC heights by altering the water vapor at the top of the PMC. Similarly, whether planetary waves and tides can also influence PMC formation through PMC heights should be investigated in future studies.

The CMN scenario also provides a new pathway to understand the response of PMCs to solar activity. In particular, the solar activity can affect the ice nucleation in PMCs by influencing the charging process of MSPs. On the one hand, the IMF $B_y$ can induce vertical electric fields in the polar ionosphere, modulating the concentration of electrons and charged-MSPs in



the upper mesosphere, ultimately affecting the ice particle radius and the PMC brightness. On the other hand, the solar UV can also modulate the concentration of charged-MSPs through ionization, but with a shorter time lag than the effects of

heating and photolysis. In order to extract solar signals in PMCs, future work should take into account both the effects of IMF $B_y$ and solar UV on the charged-MSPs.

In the future, a PMC model incorporating the CMN scheme should be developed to improve the microphysical processes, as opposed to the GS scheme applied in the majority of current PMC models. To this end, a thorough investigation of the distribution and charging process of MSPs is required, and the effects of atmospheric dynamics on PMC

height and solar activity on the charging process of MSPs should be taken into account.

*Data Availability*. The PMC data from SOFIE/AIM are available from the SOFIE website (http://sofie.gats-inc.com/sofie/index.php) and the PMC data from CIPS/AIM are available from the Laboratory for Atmospheric and Space Physics website (http://lasp.colorado.edu/aim/download/pmc/l2).

*Author Contributions*. Liang Zhang, Zhongfang Liu, and Brian Tinsley conceived the idea together. Liang Zhang analyzed

the data and drafted the manuscript. Zhongfang Liu and Brian Tinsley revised the paper.

*Competing Interests*. The authors declare that they have no conflict of interests.

*Acknowledgements*. This work was supported by the National Natural Science Foundation of China (42025602 and 41905059). We are especially grateful to the entire AIM program for providing us the continuous SOFIE and CIPS data.

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



**Table 1.** The mean values of the PMC top ($Z_{top}$), PMC maximum ($Z_{max}$), PMC bottom ($Z_{bot}$), PMC height ($h$), PMC thickness
($\Delta Z$), ice particle radius ($r$), ice particle concentration ($N$), and ice water content ($IWC$), as well as the standard deviations for
the anomalous values after removing the 35-day running mean. The PMC data are observed by SOFIE/AIM at about 70°
latitude.

|  | $Z_{top}$ | $Z_{max}$ | $Z_{bot}$ | $h$ | $\Delta Z$ | $r$ | $N$ | $IWC$ |
|---|---|---|---|---|---|---|---|---|
| SH | 86.6 km | 84.5 km | 82.5 km | 84.5 km | 4.2 km | 25.2 nm | 296 cm$^{-3}$ | 28.5 µg/m$^2$ |
| NH | 86.5 km | 83.5 km | 80.8 km | 83.7 km | 5.7 km | 27.0 nm | 317 cm$^{-3}$ | 57.5 µg/m$^2$ |
|  | $\delta Z_{top}$ | $\delta Z_{max}$ | $\delta Z_{bot}$ | $\delta h$ | $\delta \Delta Z$ | $\delta r$ | $\delta N$ | $\delta IWC$ |
| SH | 0.89 km | 0.81 km | 1.0 km | 0.84 km | 0.95 km | 5.6 nm | 242 cm$^{-3}$ | 12.8 µg/m$^2$ |
| NH | 0.65 km | 0.54 km | 0.82 km | 0.59 km | 0.89 km | 3.6 nm | 182 cm$^{-3}$ | 17.8 µg/m$^2$ |



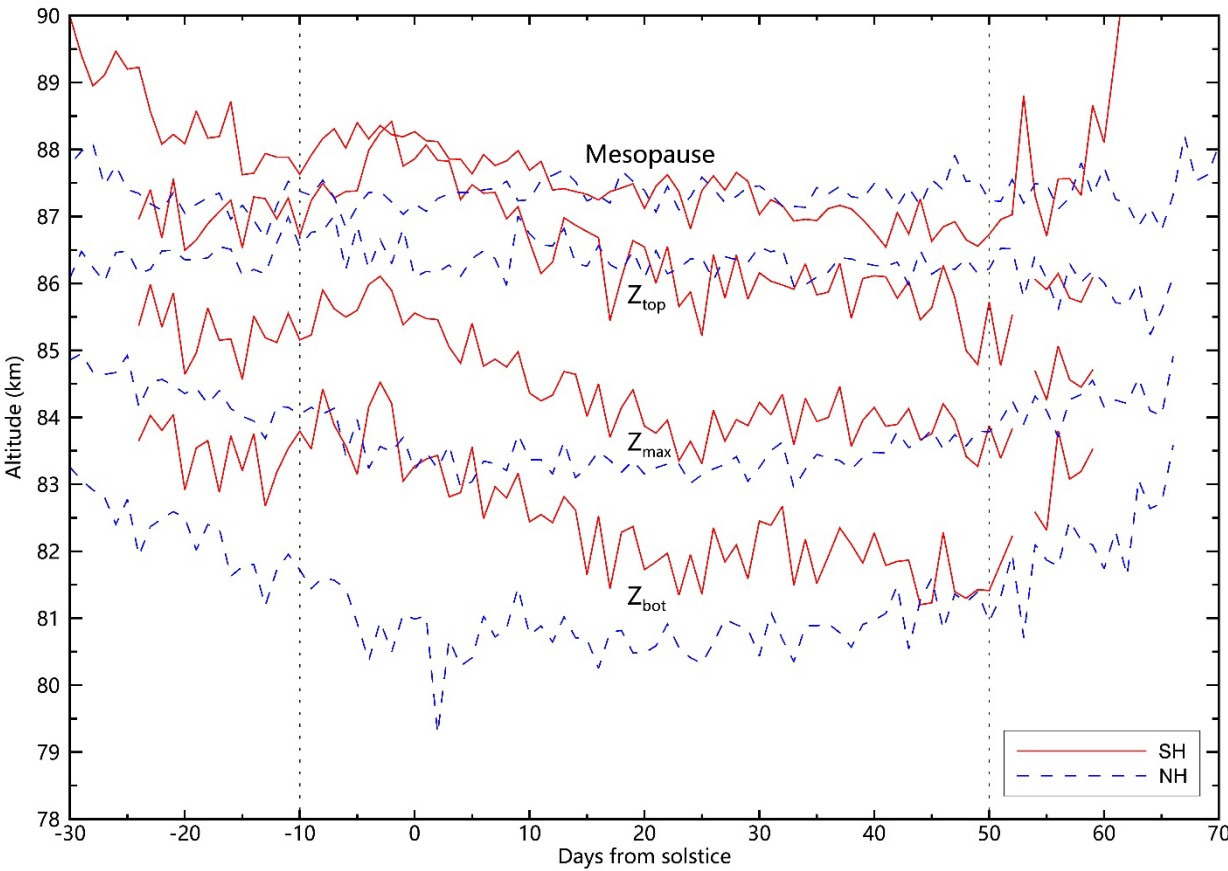

**Figure 1.** The altitudes of the top ($Z_{top}$), maximum ($Z_{max}$) and bottom ($Z_{bot}$) of the ice layer in PMCs, as well as the altitude of the mesopause. The PMC data are observed by the SOFIE/AIM at about 70° latitude.



**Figure 2.** Left panels show the anomaly of the mean ice particle radius (*r*) and PMC height (*h*) in the NH from 2007 through 2014. Right panels show the corresponding correlation coefficients for the time span of -10 to 50 days from the solstice. The PMC data are observed by the SOFIE/AIM at about 70° latitude, and the 35-day running mean are removed.







**Figure 3.** Similar to Fig. 2, but for that of the SH.





**Figure 4.** Similar to Fig. 2, but for that of the mean ice particle concentration (*N*).





**Figure 5.** Similar to Fig. 4, but for that of the SH.





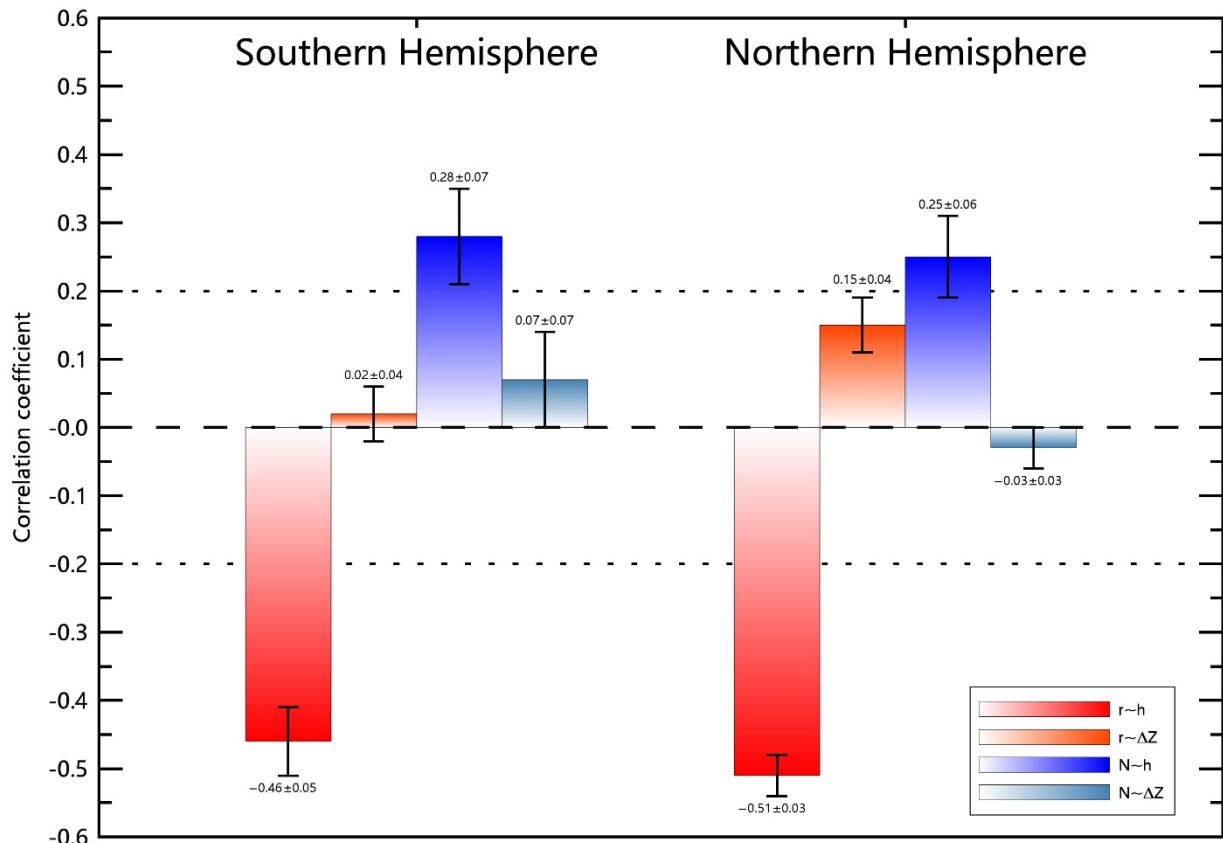

**Figure 6.** The correlation coefficients and the sensitivity for the mean ice particle radius ($r$) and concentration ($N$) with the PMC height ($h$) and thickness ($\Delta Z$) for both hemispheres. The anomaly of the PMC data is obtained by removing the 35-day running mean. The PMC data are observed by the SOFIE/AIM at about 70° latitude.



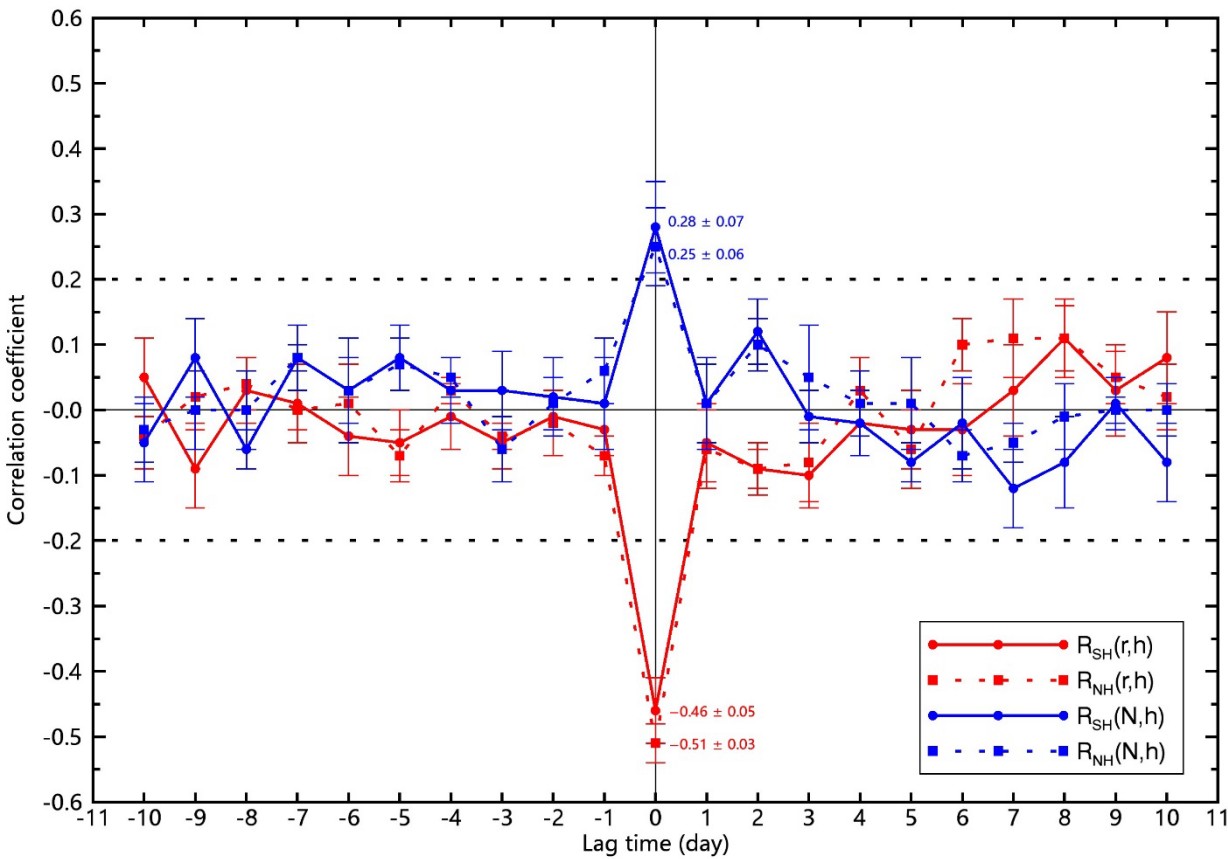

**Figure 7.** The correlation coefficients for the mean ice particle radius ($r$) and concentration ($N$) with the PMC height ($h$) for both hemispheres, with lag time varying from -10 to 10 days. The error bars illustrate the standard deviation of the means.



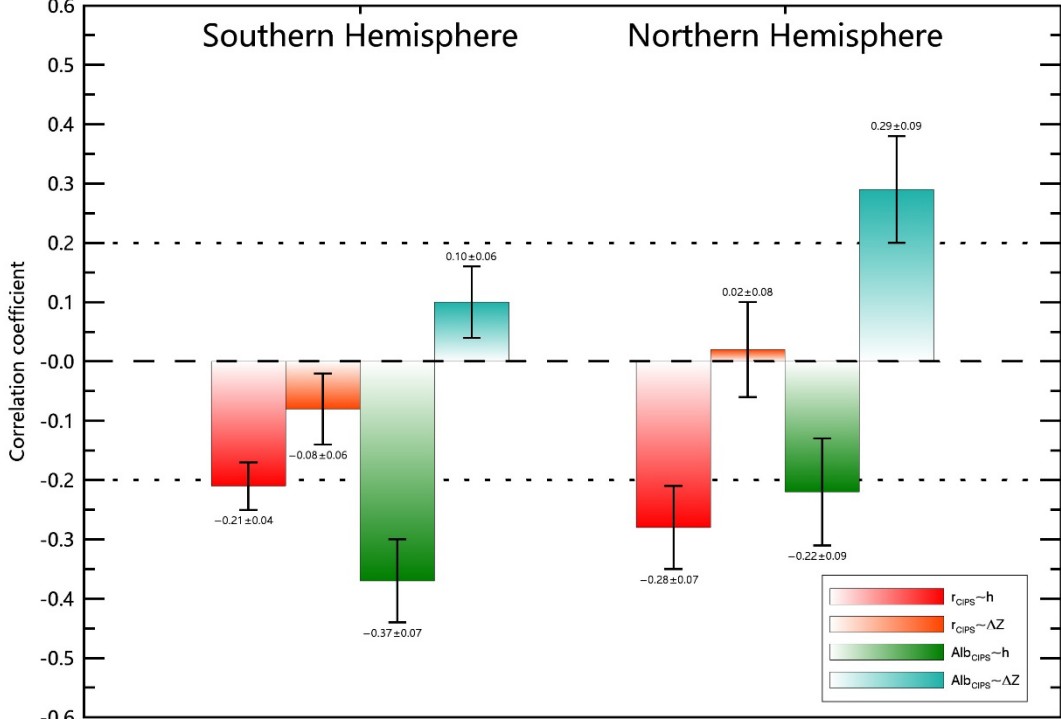

**Figure 8.** The correlation coefficients for the mean ice particle radius ($r_{CIPS}$) and albedo ($Alb_{CIPS}$) observed by CIPS/AIM at latitudes ranging from 65° to 75°, with the PMC height ($h$) and thickness ($\Delta Z$) observed by the SOFIE/AIM at about 70° latitude for both hemispheres.





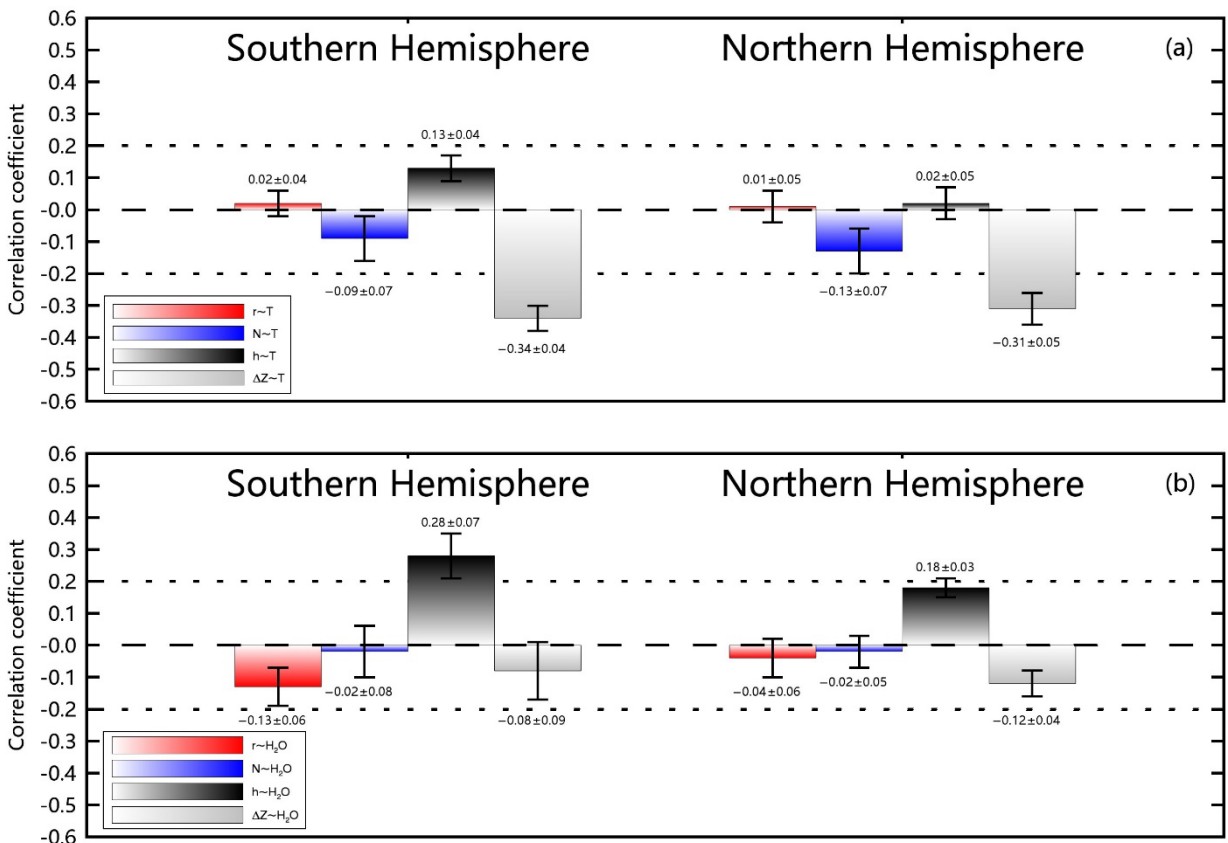

**Figure 9.** The correlation coefficients for the mean ice particle radius ($r$), concentration ($N$), PMC height ($h$), thickness ($\Delta Z$) with the mean temperature ($T$) and water vapor ($H_2O$) between 78 km and 88 km. The PMC data are observed by the SOFIE/AIM at about 70° latitude.



**Figure 10.** The correlation coefficients for the altitudes of PMC top ($Z_{top}$) and bottom ($Z_{bot}$) with the temperature and water vapor at varying altitudes.





580

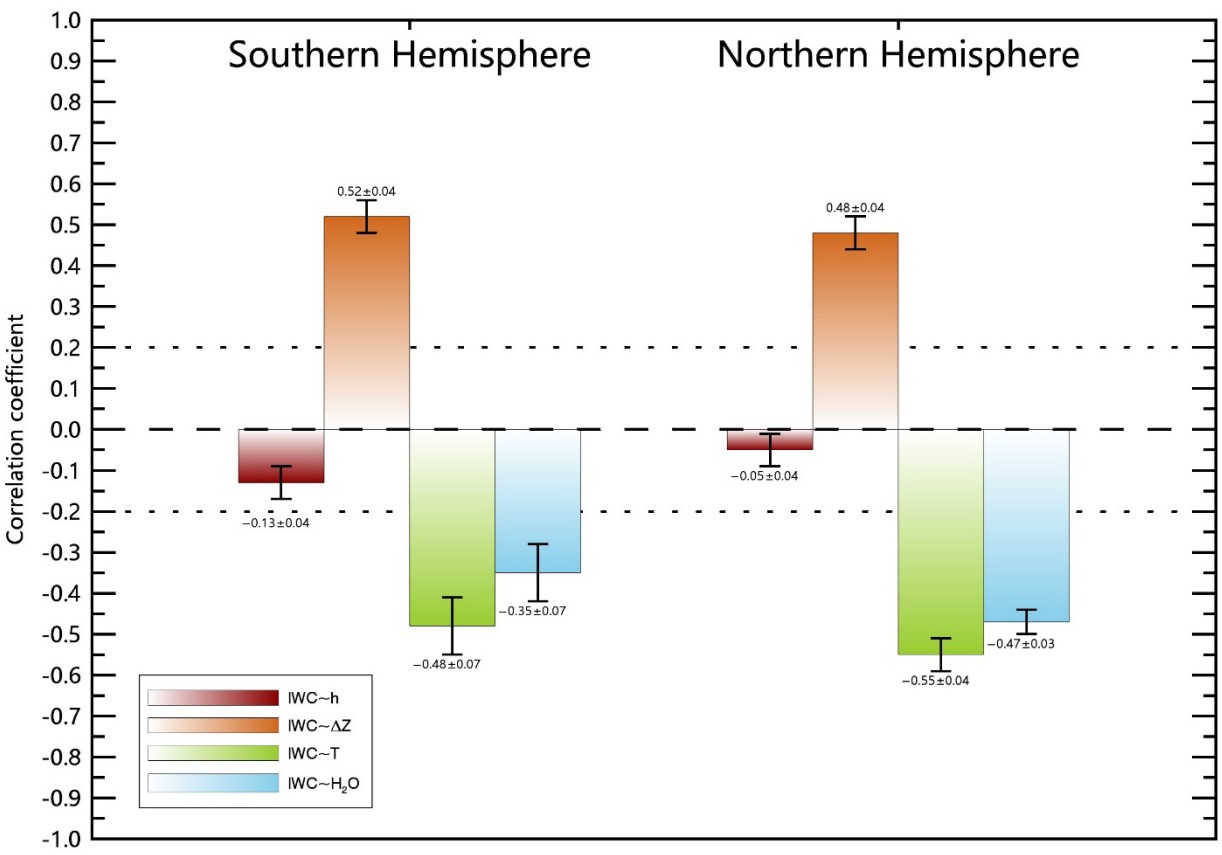

**Figure 11.** The correlation coefficients for the ice water content (*IWC*) with the PMC height (*h*), thickness (Δ*Z*), the mean temperature (*T*) and water vapor (*H$_2$O*) between 78 km and 88 km. The PMC data are observed by the SOFIE/AIM at about 70° latitude.

585



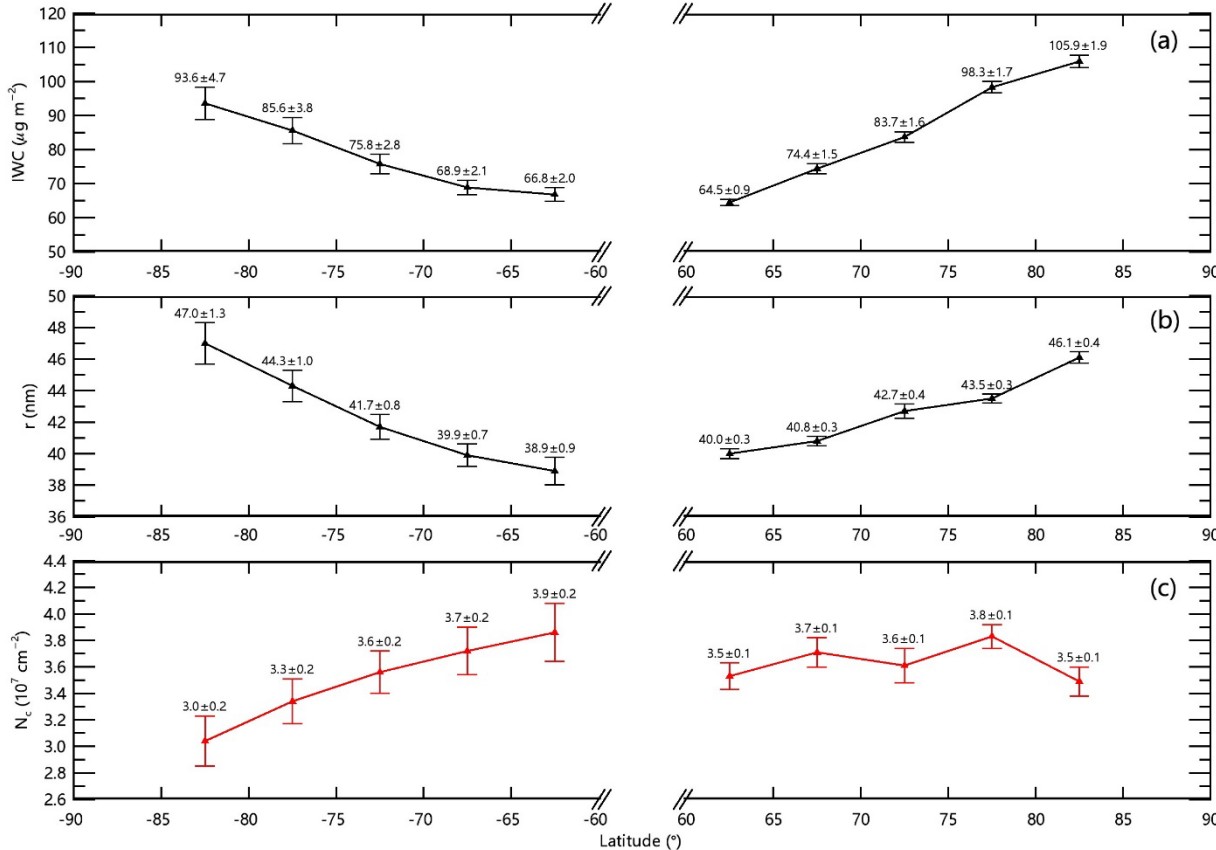

**Figure 12.** The variations of ice water content (*IWC*), ice particle radius (*r*), and ice particle column concentration ($N_c$) with latitude for both hemispheres. The PMC data are observed by the CIPS/AIM.



**Figure 13.** A schematic of the charged-MSPs nucleation (CMN) scheme for the PMC formation.







**Figure 14.** The correlation coefficients between the mean ice particle radius (*r*) and concentration (*N*) for altitudes between 79 km and 87 km for both hemispheres. The PMC data are observed by the SOFIE/AIM at about 70° latitude.



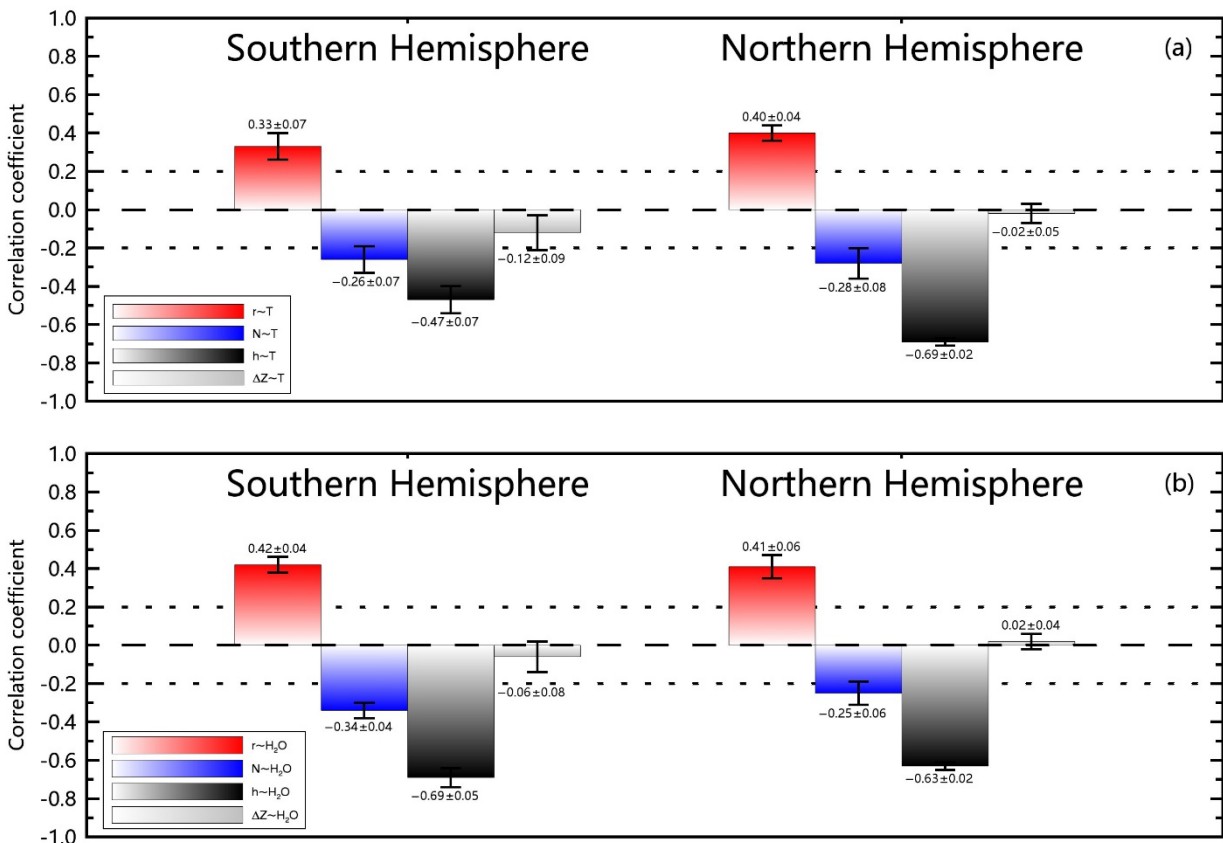

**Figure 15.** Similar to Fig. 9, but the daily mean temperature ($T$) and water vapor ($H_2O$) are averaged between the daily mean $Z_{bot}$ and $Z_{top}$, rather than between 78 km and 88 km. The PMC data are observed by the SOFIE/AIM at about 70° latitude.