# Peer review of "Dominant role of charged meteoric smoke particles in the polar mesospheric clouds"

_EGUsphere, 2024_

## Referee Comment (RC1)

Review of "Dominant role of charged meteoric smoke particles in the polar mesospheric clouds" by Zhang et al.

This article investigates the dependence of PMCs on various environmental parameters, using observations from the SOFIE and CIPS satellite instruments. The article is generally well written, but the scientific analysis is not rigorous and suffers from major flaws. As a result, the conclusions are unsupported and misleading, and I recommend that this paper is rejected.

The first major flaw is the unsupported conclusion that charged meteoric smoke particles (MSP) explain the observed variability in PMCs (even evoking this in the title). In fact, the first sentence of the conclusion states that "This study demonstrates that the charged-MSPs nucleation scheme can explain a number of puzzles in the growth-sedimentation scheme and is likely to be responsible for the formation of PMCs". This connection is made without any rigorous analysis, but instead is supported with loose associations and conjecture. Making such a bold statement requires clear evidence from the observations, or clear indications from a theoretical approach. The paper offers neither. The idea of charged MSP as PMC nuclei has been examine by previous authors (referenced in this paper), who performed rigorous model experiments and considered observations. Advancing upon these studies will require much more than the Authors have performed for this paper. The Authors furthermore ignored the SOFIE observations of the MSP content in PMC ice particles, which are reported vs. height. These unique measurements are surely relevant to an investigation of how MSPs affect PMCs, and should be considered here.

The second major flaw is the conclusion that the primary influence on PMC characteristics is altitude. This is somewhat ridiculous, as altitude is simply a coordinate, and not a forcing variable. This idea ignores the fact that PMCs are influenced by a variety of environmental parameters including temperature, water vapor, pressure, vertical wind, MSP, etc…, all of which vary in height, and act together to influence PMCs. It would be irresponsible and damaging to publish such simplistic ideas in the scientific literature.

The third major flaw is that the Authors do not appear to understand the current benchmark PMC models (e.g., WACCM-CARMA, LIMA-MIMAS), although most of these are discussed by papers in the reference list. They fail to recognize that existing models do a very good job of explaining the horizontal, vertical, and time dependence observed in PMCs [e.g., Bardeen et al., 2010; Kuilman et al., 2017; Wilms et al., 2016, and others]. Because the current state of model-observation agreement was not examined, it is difficult to understand the magnitude of the problem at hand (i.e., the effect of nucleation on PMCs).

Despite these flaws, the subject matter is of interest and the data used is of high quality, and I believe this investigation could represent a useful contribution after a major effort and restructuring.

**Specific Comments**

Placing all of the figures are all at the end of the paper makes the review unnecessarily tedious. It is now common to locate the figures inline, after the associated text.

Line 29: There is a more recent reference on space traffic effects on PMC [Stevens et al., 2022].

Line 41: The debate is not really "intense"

Line 43: It is not fair to state that nucleation is the most significant uncertainty. Rather, there are several references in this paper which suggest that nucleation is not important when describing PMC variability [Megner et al., 2011; Hervig et al., 2009c]. Even more curious is that Wilms et al. [2016] performed detailed model studies and conclude that "low MSP number density (or low nucleation rate per particle) is not a hindrance of NLC development; it is rather a prerequisite." In any case, the present study needs to do a much better job of documenting the current state of our understanding.

Line 59: State the typical PMC altitudes here.

Line 71: "sublimate"

Line 85: This statement is not supported by observations, as there are no global or routine observations of MSP size.

Line 91: A better reference here would be Baumgarten et al. [2012] (there may be a more recent paper, please check).

Line 112: On the SOFIE webpage, it looks like SOFIE reports the amount of MSPs contained in PMC particles. This quantity would be highly relevant to your study, and you should at least take a look at the measurements. Also, all of the SOFIE PMC retrievals are reported vs. height. You should mention these details here.

Line 115: On the SOFIE web page it looks like there are NH PMC observations also in 2015 and 2020-2022.

Line 117: Delete "rectangular"

Line 119: On the CIPS webpage, CIPS PMC data are available through 2022 in the NH and through 2023 in the SH.

Line 129: The section title "Effects of Altitude" is inappropriate, since altitude itself does not modulate PMC properties but rather is simply a coordinate.

Figures 2 - 5: What altitude are these results for? Is it for Zmax or something else?

Lines 135-140: These statements are not supported by the present results in any way, and in fact are somewhat nonsensical. Please see the model-SOFIE comparisons in Bardeen et al. [2010], which indicate that WACCM-CARMA simulations are in very good agreement with the observed height dependence in r and N. This is a strong indication that the microphysics in current (GS) models is fundamentally correct. The statement that r and N do not vary with height in the current GS scheme (models) is incorrect and not supported.

Line 146: Note that CIPS can not determine the PMC height, and only reports IWC which is a vertical integral. Since CIPS does not have the ability to examine height dependence in any way, this statement is nonsense.

Line 188: Your statement here could be easily tested with the thermodynamic equilibrium approach described a few lines later. SOFIE provides everything you would need (T, H2O, P), and comparing the simulations to the measured Qice would be a tangible indication of

microphysical (vs. thermodynamic) influences. This type of analysis could elevate the paper from conjecture to quantification.

Line 200: There are references on the nature of MSP particles this that should be quoted here. The Megner et al. papers are in your reference list, but you should also look at Bardeen et al. [2010]. Note that these are model simulations, and that the only observations are from a few rocket experiments [e.g., Havnes et al., 2019]. These papers will quantify how much the MSP N and r vary with height, and these details should be considered here, rather than just broad speculation.

Line 253, Fig 7: You examine the PMC r and N vs. height, all of which are observed simultaneously (i.e., the PMC properties are measured at each height in a cloud). Why then would you search for a time lag between the height and N (or r)? This seems like nonsense, and it would be surprising if a time lag was actually discovered.

Line 270: To state that this is "predicted by the CMN scheme" is unfair, since the paper does not present observations or simulations of the effect, but rather only makes speculative connections.

Line 295: The conclusions section contains an incredible number of unsupported statements, and publishing this would be irresponsible and damaging to the scientific literature.

---

## Referee Comment (RC2)

Review of egusphere-2024-1259 "Dominant role of charged meteoric smoke particles in the polar mesospheric clouds" by Liang Zhang, Zhongfang Liu, and Brian Tinsley

This manuscript is about the potential role of charged meteoric smoke particles (MSP) as condensation nuclei for polar mesospheric clouds (PMC). This is an important issue. Current models of PMC mostly consider nucleation on neutral MSP as nucleation process. As compared to nucleation on neutral MSP, nucleation on charged MSP can indeed be expected to lead to rather different nucleation rates and to a rather different altitude distribution of nucleation events. And indeed, there is today substantial observational and theoretical evidence that a substantial fraction of MSP in the mesosphere is charged. I consider the relative important of nucleation on neutral MSP and on charged MSP as an important open question for understanding PMC. So I fully agree with the authors that the importance and the consequences of "charged MSP nucleation (CMN)" is an important topic to investigate.

Then, unfortunately, I find that the authors put this topic in the wrong perspective. Throughout the manuscript, they contrast PMC nucleation on charged smoke to the "growth and sedimentation" scheme that is commonly used when describing the growth of PMC ice particles. This entire starting point for the manuscript is not meaningful.

We cannot really doubt that there are basically three distinct stages in the evolution of a PMC ice particle: (1) an initial ice particle nucleation, (2) ice particle growth in a supersaturated environment, and (3) sublimation of the ice particle when it encounters an unsaturated environment. The question of the nucleation mechanism e.g. by charged MSPs belongs to point (1). Once nucleation has occurred, stage (2) takes over and PMC ice particles will grow through deposition of water vapour as long they are in a supersaturated environment. The growth rate is essentially determined by the surrounding number density of water vapour (e.g., Hesstvedt, J. Geophys. Res., 1961). While they exist, the ice particles will of course be subject to gravity, and hence they will sediment relative to the surrounding air. Both the growth and sedimentation are basic physics, not mere assumptions introduced by a "growth and sedimentation model". They do take place in stage (2), independent of the question whether the nucleation in stage (1) occurred because of charged MSP, neutral MSP or any other nucleation process.

So contrasting "charged MSP nucleation (CMN)" and "growth-sedimentation (GS)" does not make sense. The manuscript would have made perfect sense if it instead contrasted "charged MSP nucleation (CMN)" and "neutral MSP nucleation (NMN)" within stage (1). As opposed to growth and sedimentation, the nature of the nucleation process can indeed be regarded as an "assumption" in current PMC models, as long as these models do not explicitly simulate what fraction of the MSPs is charged and what fraction of the nucleation events takes place on charged MSPs. So, I would consider this a very interesting manuscript if it contrasted CMN and NMN, and if it investigated the consequences that either scenario has on the resulting PMC properties and PMC lifecyle.

With its current focus on "CMN vs. GS", I do not consider this manuscript to be publishable. Now a question to me as reviewer is: Do I think that the current manuscript can be revised to a form that is acceptable for publication. Unfortunately, my answer is no. In order to make this manuscript publishable, it would not only be necessary to give it a new focus on "CMN vs. NMN", as outlined above. It would also be necessary to handle things much more rigorously. The manuscript contains good ideas concerning the properties of PMC resulting from nucleation on charged MSPs. And the manuscript also contains a valuable collection of statistical studies of AIM/SOFIE and AIM/CIPS satellite data. However, neither the connections made between charged MSP nucleation and resulting PMC properties, nor the subsequent connections to the AIM data are rigorous enough. There is too much hand-waving. I think that in order to make this work rigorous, one really needs to

involve a microphysical PMC model of suitable complexity. Model simulations are needed to show how charged MSP nucleation really would lead to modified PMC properties as suggested by the authors.

However, this is much work and beyond the scope of a revised manuscript. The ideas I have outlined above essentially refer to a different paper. Hence, from an ACP perspective, I suggest to reject the current manuscript, and then possibly to hope for a new one.

Below, I add some more comments that may be useful for the continued work.

When referring to PMC trends in section 1.1, it would be good to refer to the latest publications (DeLand and Thomas, Atmos. Chem. Phys., 19, 7913–7925, doi: 10.5194/acp-19-7913-2019, 2019). Also, "global warming" (or global cooling) is the wrong term when referring to the effect of a methane trend on PMC. Methane primarily affects PMCs in terms of water vapour.

The observed connection between IMF and PMC is intriguing. However, people have looked for a connection between PMC and mesospheric charging conditions in many different ways. Most importantly, there is no real evidence that PMC occurrence would be affected by geomagnetic activity. This argues against a very direct connection between charging and nucleation. For the manuscript, it would be good to look deeper into different aspects of this.

In line 76, the authors state "it seems unreasonable that ice nucleation takes place primarily at the top of the PMC, since the saturation levels throughout the PMC altitude range all favour PMC formation". This statement ignores the fact that relative humidities much larger than 100% are needed for ice nucleation on tiny MSPs (curvature effect, Kelvin equation).

The negative dependence of particle radius on PMC height in figures 2+3 is not at all surprising. It is consistent with the growth/sedimentation scheme. What is missing in the manuscript is a clear statement that the particle growth rate essentially is controlled by the absolute number density of water vapour (not the water vapour mixing ratio). The absolute water vapour number density decreases quickly with altitude, both because of the total atmospheric density decrease and because of the efficient photolysis of water vapour in the upper summer mesosphere. Hence, PMC particle growth rates decrease quickly with altitude. This fact removes many of the "growth/sedimentation inconsistencies" claimed by the authors, e.g. figues 2+3, 4+5 etc.

When discussion the effect of (wave) dynamics on PMCs, one really needs to consider the time scales. As the authors state, most current models suggest that it takes an ice particle many hours to grow to visible size. However, once a particle enters an unsaturated region (e.g. because of wave activity) sublimation to sub-visible sizes can happen on much shorter time scales than an hour. So seeing wave structures in PMCs does not contradict current descriptions of the PMC physics. Again, the growth process and the time scales involved are all basic physics (e.g. Hesstvedt, 1961). As pointed out above, Invoking an appropriate growth description (in a microphysical model) will be needed as an important step towards making the authors ideas more quantitative.

Indeed, there are many observations of dual layer or multi-layer PMCs. These do not contradict the basic idea of the growth/sedimentation process. The PMC lifecycle is not a one-dimensional process. There are strong horizontal winds near the mesopause and, more importantly, there are substantial wind shears. Hence, structures in PMC can never be understood based microphysics alone. (Still, a one-dimensional microphysical model may be sufficient to better quantify many of the ideas brought forward in the manuscript.)

The manuscript refers much to correlations and anti-correlations of various PMC properties with water vapour. The manuscript also correctly refers to the importance of "freeze-drying". The effect of freeze-drying really makes any correlation discussions tricky. You very often run into a chicken-and-egg problem, depending on the concrete atmospheric situation and history.

Maybe I misunderstand the time-lag analysis in figure 7. For me, the method does not make sense. The authors correlate particle radius and particle concentration inferred from AIM/CIPS with the cloud height inferred from AIM/SOFIE. As expected, there is correlation/anticorrelation. But of course, these correlations completely disappear when correlating the AIM data with SOFIE data from a different day. On different days, the instruments observe different clouds, so of course there will not be any correlation.

An anti-correlation of PMC brightness and cloud radius as suggested by Rusch et al. (2017) (line281) is problematic. CIPS infers a kind of "column-averaged" mean particle radius. This quantity is usually useful, but becomes ill-defined in cases you have an aged PMC at the end of its lifecycle with large particles near the cloud bottom and nothing left above. The conclusions of Rusch et al. (2017) are likely affected by this, as discussed e.g. by Hultgren and Gumbel (2014).

The idea that nucleation not necessarily happens at the mesopause temperature minimum is not new. A nice reference is "Berger, U., and U. von Zahn (2007), Three-dimensional modeling of the trajectories of visible noctilucent cloud particles: An indication of particle nucleation well below the mesopause, J. Geophys. Res., 112, D16204, doi:10.1029/2006JD008106." These authors do not start out from a nucleation process. Vice versa, they trace the particles in the visible part of the cloud back to where the nucleation must have taken place. They suggest that nucleation in average takes place 3 km above the visible cloud.

---

## Author Comment (AC1)

**Response to Referee #1 Comments on egusphere-2024-1259, "Dominant role charged meteoric smoke particles in the polar mesospheric clouds"**

We thank the Referee #1 for the valuable and constructive comments. The following are the point-by-point responses.

This article investigates the dependence of PMCs on various environmental parameters, using observations from the SOFIE and CIPS satellite instruments. The article is generally well written, but the scientific analysis is not rigorous and suffers from major flaws. As a result, the conclusions are unsupported and misleading, and I recommend that this paper is rejected.

We are grateful to the referee for providing so many valuable suggestions, although the decision is to reject this manuscript.

The first major flaw is the unsupported conclusion that charged meteoric smoke particles (MSP) explain the observed variability in PMCs (even evoking this in the title). In fact, the first sentence of the conclusion states that "This study demonstrates that the charged-MSPs nucleation scheme can explain a number of puzzles in the growth-sedimentation scheme and is likely to be responsible for the formation of PMCs". This connection is made without any rigorous analysis, but instead is supported with loose associations and conjecture. Making such a bold statement requires clear evidence from the observations, or clear indications from a theoretical approach. The paper offers neither. The idea of charged MSP as PMC nuclei has been examine by previous authors (referenced in this paper), who performed rigorous model experiments and considered observations. Advancing upon these studies will require much more than the Authors have performed for this paper. The Authors furthermore ignored the SOFIE observations of the MSP content in PMC ice particles, which are reported vs. height. These unique measurements are surely relevant to an investigation of how MSPs affect PMCs, and should be considered here.

By analyzing the SOFIE data, we obtained the results shown in Figs. 2-6, i.e. the averaged ice particle concentration and radius in PMCs are significantly correlated with the mean PMC heights. Our explanation for these observations is that: since the electron density increases exponentially with altitude, and more charged-MSPs act as ice nuclei when PMCs occur at higher altitudes, the mean ice particle concentration and radius are sensitive to the mean PMC heights.

Our previous work has provided observational evidence that charged-MSPs can act as ice nuclei. The By component of the solar wind magnetic field (IMF_By) can affect the electric field and the electron density in the mesosphere, and we found that the IMF_By is significantly correlated with the ice particle radius in PMCs. Based on the above work, this paper further shows that the altitude distribution of charged-MSPs has a more important effect on PMCs.

The second major flaw is the conclusion that the primary influence on PMC characteristics is altitude. This is somewhat ridiculous, as altitude is simply a coordinate, and not a forcing variable. This idea ignores the fact that PMCs are influenced by a variety of environmental parameters including temperature, water

vapor, pressure, vertical wind, MSP, etc…, all of which vary in height, and act together to influence PMCs. It would be irresponsible and damaging to publish such simplistic ideas in the scientific literature.

> We apologize for the language problems that lead to this ambiguity.
>
> The "altitude" used in the manuscript is the mean PMC height, $h=(Z_{top}+Z_{bot})/2$, which varies from day to day. The electron density varies exponentially with altitude, so the variation in the mean PMC height will affect the concentration of charged-MSPs, and further affect the column mean ice particle concentration and radius.
>
> The effects of water vapor and temperature with altitude have also been investigated in the work. We find that the mean temperature mainly affects the IWC (Fig. 11), but does not affect the column mean ice particle concentration and radius (Fig. 9). Diffusion and vertical wind are important for ice particle transport, but are beyond the scope of this work.
>
> In addition, as shown in Figs. 2-5 and Table 1, the standard deviation of the mean PMC height ($\delta h$) is only about 0.8 km in SH and 0.5 km in NH. For such a small $\delta h$, the environment parameters such as vertical wind, diffusion, MSP, etc. are not expected to vary significantly. However, the electron density varies exponentially with altitude, thus the charged-MSPs are expected to be more sensitive to the small $\delta h$.

The third major flaw is that the Authors do not appear to understand the current benchmark PMC models (e.g., WACCM-CARMA, LIMA-MIMAS), although most of these are discussed by papers in the reference list. They fail to recognize that existing models do a very good job of explaining the horizontal, vertical, and time dependence observed in PMCs [e.g., Bardeen et al., 2010; Kuilman et al., 2017; Wilms et al., 2016, and others]. Because the current state of model-observation agreement was not examined, it is difficult to understand the magnitude of the problem at hand (i.e., the effect of nucleation on PMCs).

> In fact, the main difference between the CMN and GS schemes is the initial distribution of ice nuclei. In the CMN scheme, the distribution of ice nuclei is assumed to be determined by the vertical distribution of electrons, and the sedimentation is not necessary. The CMN scheme does not contradict with the results of the CARMA or MIMAS models. Our intention is to try to improve these PMC models by reducing the computational time, rather than to deny their achievements.

Despite these flaws, the subject matter is of interest and the data used is of high quality, and I believe this investigation could represent a useful contribution after a major effort and restructuring.

> We are very grateful for this comment.

**Specific Comments**
Placing all of the figures are all at the end of the paper makes the review unnecessarily tedious. It is now common to locate the figures inline, after the associated text.
Line 29: There is a more recent reference on space traffic effects on PMC [Stevens et al., 2022].

Line 41: The debate is not really "intense".

    Agreed. Thanks.

Line 43: It is not fair to state that nucleation is the most significant uncertainty. Rather, there are several references in this paper which suggest that nucleation is not important when describing PMC variability [Megner et al., 2011; Hervig et al., 2009c]. Even more curious is that Wilms et al. [2016] performed detailed model studies and conclude that "low MSP number density (or low nucleation rate per particle) is not a hindrance of NLC development; it is rather a prerequisite." In any case, the present study needs to do a much better job of documenting the current state of our understanding.

    In the CMN scheme, the IWC is determined by the environment temperature rather than nucleation, so we fully agree with the referee that the nucleation is not important for the variability of IWC and occurrence of PMCs. On the other hand, the column mean ice particle radius and concentration are determined by the nucleation rather than temperature or water vapor, in other words, the nucleation plays an important role in the microphysical processes from the perspective of the CMN scheme.

Line 59: State the typical PMC altitudes here.
Line 71: "sublimate"
Line 85: This statement is not supported by observations, as there are no global or routine observations of MSP size.
Line 91: A better reference here would be Baumgarten et al. [2012] (there may be a more recent paper, please check).
Line 112: On the SOFIE webpage, it looks like SOFIE reports the amount of MSPs contained in PMC particles. This quantity would be highly relevant to your study, and you should at least take a look at the measurements. Also, all of the SOFIE PMC retrievals are reported vs. height. You should mention these details here.
Line 115: On the SOFIE web page it looks like there are NH PMC observations also in 2015 and 2020-2022.
Line 117: Delete "rectangular"
Line 119: On the CIPS webpage, CIPS PMC data are available through 2022 in the NH and through 2023 in the SH.

    We are agreed and thanks for these suggestions.

Line 129: The section title "Effects of Altitude" is inappropriate, since altitude itself does not modulate PMC properties but rather is simply a coordinate.
Figures 2 - 5: What altitude are these results for? Is it for Zmax or something else?

    We apologize for this ambiguity in language. The "altitude" ($h$) in Figs. 2-6 is the daily mean height of PMCs, which is calculated by $h = (Z_{top}+Z_{bot})/2$. The

Lines 135-140: These statements are not supported by the present results in any way, and in fact are somewhat nonsensical. Please see the model-SOFIE comparisons in Bardeen et al. [2010], which indicate that WACCM-CARMA simulations are in very good agreement with the observed height dependence in r and N. This is a strong indication that the microphysics in current (GS) models is fundamentally correct. The statement that r and N do not vary with height in the current GS scheme (models) is incorrect and not supported.

> The altitude $h$ is the daily mean height of PMCs, $h = (Z_{top}+Z_{bot})/2$. The $r$ is the column mean of ice particle radius, which is averaged between the daily mean $Z_{bot}$ and $Z_{top}$. The $N$ is the column mean of ice particle concentration.

> The results in Figs. 2-6 show that the column mean of $r$ and $N$ are significantly correlated with the mean height of PMCs, these results are different from the variability of ice particles inside PMCs with height in the GS scheme.

Line 146: Note that CIPS cannot determine the PMC height, and only reports IWC which is a vertical integral. Since CIPS does not have the ability to examine height dependence in any way, this statement is nonsense.

> The PMC height ($h$) in Fig. 8 was measured by SOFIE at about 69°, and the ice particle radius and albedo observed by CIPS between 65°-70° are applied.

Line 188: Your statement here could be easily tested with the thermodynamic equilibrium approach described a few lines later. SOFIE provides everything you would need (T, $H_2O$, P), and comparing the simulations to the measured $Q_{ice}$ would be a tangible indication of microphysical (vs. thermodynamic) influences. This type of analysis could elevate the paper from conjecture to quantification.

> The references of simulations based on the thermodynamic equilibrium have been cited in the manuscript. We will build a new 1-D model based on the CMN scheme to quantify our claims.

Line 200: There are references on the nature of MSP particles this that should be quoted here. The Megner et al. papers are in your reference list, but you should also look at Bardeen et al. [2010]. Note that these are model simulations, and that the only observations are from a few rocket experiments [e.g., Havnes et al., 2019]. These papers will quantify how much the MSP N and r vary with height, and these details should be considered here, rather than just broad speculation.

> We are grateful for these suggestions.

Line 253, Fig 7: You examine the PMC r and N vs. height, all of which are observed simultaneously (i.e., the PMC properties are measured at each height in a cloud). Why then would you search for a time lag between the height and N (or r)? This seems like nonsense, and it would be surprising if a time lag was actually discovered.

> From the perspective of CMN scheme, the column mean $r$ and $N$ are determined by the mean PMC height $h$, so the zero-day lag time shows that the ice particles grow rapidly.

Line 270: To state that this is "predicted by the CMN scheme" is unfair, since the paper does not present observations or simulations of the effect, but rather only makes speculative connections.

> Thanks.

Line 295: The conclusions section contains an incredible number of unsupported statements, and publishing this would be irresponsible and damaging to the scientific literature.

> Thanks.

---

## Author Comment (AC2)

**Response to Referee #2 Comments on egusphere-2024-1259, "Dominant role charged meteoric smoke particles in the polar mesospheric clouds"**

We would like to thank the Referee #2 for the valuable comments, which are very helpful for improving this article. In the following the remarks are responded point by point.

This manuscript is about the potential role of charged meteoric smoke particles (MSP) as condensation nuclei for polar mesospheric clouds (PMC). This is an important issue. Current models of PMC mostly consider nucleation on neutral MSP as nucleation process. As compared to nucleation on neutral MSP, nucleation on charged MSP can indeed be expected to lead to rather different nucleation rates and to a rather different altitude distribution of nucleation events. And indeed, there is today substantial observational and theoretical evidence that a substantial fraction of MSP in the mesosphere is charged. I consider the relative important of nucleation on neutral MSP and on charged MSP as an important open question for understanding PMC. So I fully agree with the authors that the importance and the consequences of "charged MSP nucleation (CMN)" is an important topic to investigate.

We are very grateful to the referee for supporting the importance of the CMN scheme in PMCs.

Then, unfortunately, I find that the authors put this topic in the wrong perspective. Throughout the manuscript, they contrast PMC nucleation on charged smoke to the "growth and sedimentation" scheme that is commonly used when describing the growth of PMC ice particles. This entire starting point for the manuscript is not meaningful.

Although the GS scheme is widely used and almost the only theory for the PMC formation, but we find that the "sedimentation" in GS scheme is unimportant and unnecessary. The logic of this manuscript is simple: if the GS scheme is proved to be (partly) incorrect, then the only alternative theory for PMC formation will be the CMN scheme. That's why we try to prove that CMN is correct but GS is incorrect throughout the manuscript.

We cannot really doubt that there are basically three distinct stages in the evolution of a PMC ice particle: (1) an initial ice particle nucleation, (2) ice particle growth in a supersaturated environment, and (3) sublimation of the ice particle when it encounters an unsaturated environment. The question of the nucleation mechanism e.g. by charged MSPs belongs to point (1). Once nucleation has occurred, stage (2) takes over and PMC ice particles will grow through deposition of water vapour as long they are in a supersaturated environment. The growth rate is essentially determined by the surrounding number density of water vapour (e.g., Hesstvedt, J. Geophys. Res., 1961). While they exist, the ice particles will of course be subject to gravity, and hence they will sediment relative to the surrounding air. Both the growth and sedimentation are basic physics, not mere assumptions introduced by a "growth and sedimentation model". They do take place in stage (2), independent of the question whether the nucleation in stage (1) occurred because of charged MSP, neutral MSP or any other nucleation process.
So contrasting "charged MSP nucleation (CMN)" and "growth-sedimentation (GS)" does not make sense. The manuscript would have made perfect sense if it instead contrasted "charged MSP nucleation (CMN)" and "neutral MSP nucleation (NMN)"

within stage (1). As opposed to growth and sedimentation, the nature of the nucleation process can indeed be regarded as an "assumption" in current PMC models, as long as these models do not explicitly simulate what fraction of the MSPs is charged and what fraction of the nucleation events takes place on charged MSPs. So, I would consider this a very interesting manuscript if it contrasted CMN and NMN, and if it investigated the consequences that either scenario has on the resulting PMC properties and PMC lifecyle.

We are in favor of the "nucleation" and "growth" processes, but we strongly doubt the importance of the "sedimentation" processes. Sedimentation of ice particles affected by gravity and upwelling is too slow to explain the rapid variability of PMCs.

The major difference between the CMN and the GS process is that the CMN can explain the observed PMC phenomenon in the absence of "sedimentation". In other words, there is no need to simulate particle trajectories in the CMN model, which significantly reduces the computational time.

With its current focus on "CMN vs. GS", I do not consider this manuscript to be publishable. Now a question to me as reviewer is: Do I think that the current manuscript can be revised to a form that is acceptable for publication. Unfortunately, my answer is no. In order to make this manuscript publishable, it would not only be necessary to give it a new focus on "CMN vs. NM", as outlined above. It would also be necessary to handle things much more rigorously. The manuscript contains good ideas concerning the properties of PMC resulting from nucleation on charged MSPs. And the manuscript also contains a valuable collection of statistical studies of AIM/SOFIE and AIM/CIPS satellite data. However, neither the connections made between charged MSP nucleation and resulting PMC properties, nor the subsequent connections to the AIM data are rigorous enough. There is too much hand-waving. I think that in order to make this work rigorous, one really needs to involve a microphysical PMC model of suitable complexity. Model simulations are needed to show how charged MSP nucleation really would lead to modified PMC properties as suggested by the authors.
However, this is much work and beyond the scope of a revised manuscript. The ideas I have outlined above essentially refer to a different paper. Hence, from an ACP perspective, I suggest to reject the current manuscript, and then possibly to hope for a new one.

We respect the referee's decision to reject this manuscript, and we are very grateful for the valuable advice. We strongly agree that a new PMC model should be developed to support our hypothesis. We will prepare and submit a better manuscript in the future.

Below, I add some more comments that may be useful for the continued work.

When referring to PMC trends in section 1.1, it would be good to refer to the latest publications (DeLand and Thomas, Atmos. Chem. Phys., 19, 7913−7925, doi: 10.5194/acp-19-7913-2019, 2019). Also, "global warming" (or global cooling) is the wrong term when referring to the effect of a methane trend on PMC. Methane primarily affects PMCs in terms of water vapour.

Agreed. Thanks.

The observed connection between IMF and PMC is intriguing. However, people have looked for a connection between PMC and mesospheric charging conditions in many different ways. Most importantly, there is no real evidence that PMC occurrence would be affected by geomagnetic activity. This argues against a very direct connection between charging and nucleation. For the manuscript, it would be good to look deeper into different aspects of this.

By analyzing the SOFIE PMC data, we also found no signals of geomagnetic or lightning activity. However, we did find a weak link between IMF $B_y$ and PMCs, and we are verifying the significance of this link by analyzing data from other satellites.

It should be noted that, according to the CMN scheme, the IWC of PMC is dominated by the temperature rather than the concentration of ice nuclei, as shown in Fig. 11. From the perspective of the CMN scenario, the occurrence of PMC is not expected to be influenced by the geomagnetic activity via the concentration of charged-MSPs.

In line 76, the authors state "it seems unreasonable that ice nucleation takes place primarily at the top of the PMC, since the saturation levels throughout the PMC altitude range all favour PMC formation". This statement ignores the fact that relative humidities much larger than 100% are needed for ice nucleation on tiny MSPs (curvature effect, Kelvin equation).

We are grateful for this comment. In the CMN scheme assumes that the ice nucleation can occur at any altitude. It is therefore very important to check whether ice nucleation can occur at the bottom of PMCs where the humidity is not large enough.

The negative dependence of particle radius on PMC height in figures 2+3 is not at all surprising. It is consistent with the growth/sedimentation scheme. What is missing in the manuscript is a clear statement that the particle growth rate essentially is controlled by the absolute number density of water vapour (not the water vapour mixing ratio). The absolute water vapour number density decreases quickly with altitude, both because of the total atmospheric density decrease and because of the efficient photolysis of water vapour in the upper summer mesosphere. Hence, PMC particle growth rates decrease quickly with altitude. This fact removes many of the "growth/sedimentation inconsistencies" claimed by the authors, e.g. figues 2+3, 4+5 etc.

On the one hand, the variations in PMC height ($\delta h$) are small, as shown in Table 1, thus the variation in water vapor is also small and may not account for the large variations in the column mean of $r$ and $N$.

On the other hand, Fig.9 has shown that the $r$ and $N$ are not correlated with the water vapor mixing ratio (column average between $Z_{bot}$ and $Z_{top}$), similarly, it can be easily calculated that the $r$ and $N$ are also not determined by the absolute amount of water vapor.

When discussion the effect of (wave) dynamics on PMCs, one really needs to consider the time scales. As the authors state, most current models suggest that it takes an ice particle many hours to grow to visible size. However, once a particle enters an unsaturated region (e.g. because of wave activity) sublimation to sub-visible sizes can happen on much shorter time scales than an hour. So seeing wave structures

in PMCs does not contradict current descriptions of the PMC physics. Again, the growth process and the time scales involved are all basic physics (e.g. Hesstvedt, 1961). As pointed out above, Invoking an appropriate growth description (in a microphysical model) will be needed as an important step towards making the authors ideas more quantitative.

Indeed, there are many observations of dual layer or multi-layer PMCs. These do not contradict the basic idea of the growth/sedimentation process. The PMC lifecycle is not a one-dimensional process. There are strong horizontal winds near the mesopause and, more importantly, there are substantial wind shears. Hence, structures in PMC can never be understood based microphysics alone. (Still, a one-dimensional microphysical model may be sufficient to better quantify many of the ideas brought forward in the manuscript.)

> Although the observed wave structure or multi-layer structure in PMCs has been explained by the GS scheme, however, they can also be well explained by the CMN scheme. Of course, we fully agree that a new PMC model is needed to explicitly demonstrate our claims.

The manuscript refers much to correlations and anti-correlations of various PMC properties with water vapour. The manuscript also correctly refers to the importance of "freeze-drying". The effect of freeze-drying really makes any correlation discussions tricky. You very often run into a chicken-and-egg problem, depending on the concrete atmospheric situation and history.

> The "freeze-drying effect" can be derived directly from the GS scheme, and the observed dehydration/hydration of water vapor above/below PMCs is usually attributed to the growth, sedimentation, and sublimation of ice particles. However, it should be noted that almost all simulations based on the GS theory overestimate the freeze-drying effect.

> According to the CMN scheme, the "sedimentation" is unimportant, so the redistribution of water vapor with altitudes should not result from the growth, sedimentation, and sublimation of ice particles.

> We are preparing another manuscript to show that the traditional concept of "freeze-drying" based on "sedimentation" is (partly) incorrect, in which we will also argue again that the GS scheme is wrong.

Maybe I misunderstand the time-lag analysis in figure 7. For me, the method does not make sense. The authors correlate particle radius and particle concentration inferred from AIM/CIPS with the cloud height inferred from AIM/SOFIE. As expected, there is correlation/anticorrelation. But of course, these correlations completely disappear when correlating the AIM data with SOFIE data from a different day. On different days, the instruments observe different clouds, so of course there will not be any correlation.

> Assuming that the CMN scheme is correct, namely, the column mean $r$ and $N$ are determined by the mean PMC height $h$ (the charged-MSPs increase rapidly with $h$), then the zero-day lag shown in Fig. 7 indicates that the ice particle growth is rapid, within one day. The $r$, $N$, and $h$ in Fig. 7 are all observed from SOFIE. Of course, from the perspective of the GS scheme, the method in Fig. 7 is nonsense.

> In Fig. 8, we compared the *r* from CIPS with the *h* from SOFIE, just to check that it is consistent with the result in Fig. 6. More interestingly, the IWC does not depend on *h* (shown in Fig. 11), but the *albedo* from CIPS is negatively correlated with *h*, which is possibly because the *albedo* is proportional to $r^6$. In short, the results in Fig. 8 support the results in Fig. 6 (i.e., the mean *r* depends on the mean *h*).

An anti-correlation of PMC brightness and cloud radius as suggested by Rusch et al. (2017) (line281) is problematic. CIPS infers a kind of "column-averaged" mean particle radius. This quantity is usually useful, but becomes ill-defined in cases you have an aged PMC at the end of its lifecycle with large particles near the cloud bottom and nothing left above. The conclusions of Rusch et al. (2017) are likely affected by this, as discussed e.g. by Hultgren and Gumbel (2014).

> The anti-correlation between *r* and IWC seems to be a common phenomenon in gravity wave region, as shown by Gao et al. (2018). Further work should be done to justify whether it is ill-defined. Both Rusch et al. (2017) and Gao et al. (2018) explained this phenomenon by the GS scheme, however, the CMN scheme may also provide an alternative explanation through the variation of the PMC height.

The idea that nucleation not necessarily happens at the mesopause temperature minimum is not new. A nice reference is "Berger, U., and U. von Zahn (2007), Three-dimensional modeling of the trajectories of visible noctilucent cloud particles: An indication of particle nucleation well below the mesopause, J. Geophys. Res., 112, D16204, doi:10.1029/2006JD008106." These authors do not start out from a nucleation process. Vice versa, they trace the particles in the visible part of the cloud back to where the nucleation must have taken place. They suggest that nucleation in average takes place 3 km above the visible cloud.

> Thanks a lot for this reference. The simulation by Berger and von Zahn (2007) is very interesting, clearly showing the latitudinal and vertical transport of ice particles, with a transport time of 36 hours. In the CMN scheme, the nucleation is assumed to occur in situ throughout the PMC altitude range in a much shorter time, and it is the electrons rather than the neutral MSPs that determine the distribution of ice nuclei (charged-MSPs).